# Distributed Exploration in Multi-Armed Bandits

**Eshcar Hillel**
Yahoo Labs, Haifa
eshcar@yahoo-inc.com

**Zohar Karnin**
Yahoo Labs, Haifa
zkarnin@yahoo-inc.com

**Tomer Koren**[*]
Technion — Israel Inst. of Technology
tomerk@technion.ac.il

**Ronny Lempel**
Yahoo Labs, Haifa
rlempel@yahoo-inc.com

**Oren Somekh**
Yahoo Labs, Haifa
orens@yahoo-inc.com

## Abstract

We study exploration in Multi-Armed Bandits in a setting where $k$ players collaborate in order to identify an $\varepsilon$-optimal arm. Our motivation comes from recent employment of bandit algorithms in computationally intensive, large-scale applications. Our results demonstrate a non-trivial tradeoff between the number of arm pulls required by each of the players, and the amount of communication between them. In particular, our main result shows that by allowing the $k$ players to communicate *only once*, they are able to learn $\sqrt{k}$ times faster than a single player. That is, distributing learning to $k$ players gives rise to a factor $\sqrt{k}$ parallel speed-up. We complement this result with a lower bound showing this is in general the best possible. On the other extreme, we present an algorithm that achieves the ideal factor $k$ speed-up in learning performance, with communication only logarithmic in $1/\varepsilon$.

## 1 Introduction

Over the past years, multi-armed bandit (MAB) algorithms have been employed in an increasing amount of large-scale applications. MAB algorithms rank results of search engines [23, 24], choose between stories or ads to showcase on web sites [2, 8], accelerate model selection and stochastic optimization tasks [21, 22], and more. In many of these applications, the workload is simply too high to be handled by a single processor. In the web context, for example, the sheer volume of user requests and the high rate at which they arrive, require websites to use many front-end machines that run in multiple data centers. In the case of model selection tasks, a single evaluation of a certain model or configuration might require considerable computation time, so that distributing the exploration process across several nodes may result with a significant gain in performance. In this paper, we study such large-scale MAB problems in a distributed environment where learning is performed by several independent nodes that may take actions and observe rewards in parallel.

Following recent MAB literature [14, 3, 15, 18], we focus on the problem of identifying a "good" bandit arm with high confidence. In this problem, we may repeatedly choose one arm (corresponding to an action) and observe a reward drawn from a probability distribution associated with that arm. Our goal is to find an arm with an (almost) optimal expected reward, with as few arm pulls as possible (that is, minimize the *simple regret* [7]). Our objective is thus explorative in nature, and in

---

[*]Most of this work was done while the author was at Yahoo Labs, Haifa.

particular we do not mind the incurred costs or the involved regret. This is indeed the natural goal in many applications, such as in the case of model selection problems mentioned above. In our setup, a distributed strategy is evaluated by the number of arm pulls *per node* required for the task, which correlates with the parallel speed-up obtained by distributing the learning process.

We abstract a distributed MAB system as follows. In our model, there are $k$ *players* that correspond to $k$ independent machines in a cluster. The players are presented with a set of arms, with a common goal of identifying a good arm. Each player receives a stream of queries upon each it chooses an arm to pull. This stream is usually regulated by some load balancer ensuring the load is roughly divided evenly across players. To collaborate, the players may communicate with each other. We assume that the bandwidth of the underlying network is limited, so that players cannot simply share every piece of information. Also, communicating over the network might incur substantial latencies, so players should refrain from doing so as much as possible. When measuring communication of a certain multi-player protocol we consider the number of *communication rounds* it requires, where in a round of communication each player broadcasts a single message (of arbitrary size) to all other players. Round-based models are natural in distributed learning scenarios, where frameworks such as MapReduce [11] are ubiquitous.

What is the tradeoff between the learning performance of the players, and the communication between them? At one extreme, if all players broadcast to each other each and every arm reward as it is observed, they can simply simulate the decisions of a serial, optimal algorithm. However, the communication load of this strategy is of course prohibitive. At the other extreme, if the players never communicate, each will suffer the learning curve of a single player, thereby avoiding any possible speed-up the distributed system may provide. Our goal in this work is to better understand this tradeoff between inter-player communication and learning performance.

Considering the high cost of communication, perhaps the simplest and most important question that arises is how well can the players learn while keeping communication to the very minimum. More specifically, is there a non-trivial strategy by which the players can identify a "good" arm while communicating only once, at the end of the process? As we discuss later on, this is a non-trivial question. On the positive side, we present a $k$-player algorithm that attains an asymptotic parallel speed-up of $\sqrt{k}$ factor, as compared to the conventional, serial setting. In fact, our approach demonstrates how to convert virtually any serial exploration strategy to a distributed algorithm enjoying such speed-up. Ideally, one could hope for a factor $k$ speed-up in learning performance; however, we show a lower bound on the required number of pulls in this case, implying that our $\sqrt{k}$ speed-up is essentially optimal.

At the other end of the trade-off, we investigate how much communication is necessary for obtaining the ideal factor $k$ parallel speed-up. We present a $k$-player strategy achieving such speed-up, with communication only logarithmic in $1/\varepsilon$. As a corollary, we derive an algorithm that demonstrates an explicit trade-off between the number of arm pulls and the amount of inter-player communication.

## 1.1 Related Work

Recently there has been an increasing interest in distributed and collaborative learning problems. In the MAB literature, several recent works consider multi-player MAB scenarios in which players actually *compete* with each other, either on arm-pulls resources [15] or on the rewards received [19]. In contrast, we study a *collaborative* multi-player problem and investigate how sharing observations helps players achieve their common goal. The related work of Kanade et al. [17] in the context of non-stochastic (i.e. adversarial) experts also deals with a collaborative problem in a similar distributed setup, and examine the trade-off between communication and the cumulative regret.

Another line of recent work was focused on distributed stochastic optimization [13, 1, 12] and distributed PAC models [6, 10, 9], investigating the involved communication trade-offs. The techniques developed there, however, are inherently "batch" learning methods and thus are not directly applicable to our MAB problem which is online in nature. Questions involving network topology [13, 12] and delays [1] are relevant to our setup as well; however, our present work focuses on establishing the first non-trivial guarantees in a distributed collaborative MAB setting.

## 2 Problem Setup and Statement of Results

In our model of the Distributed Multi-Armed Bandit problem, there are $k \geq 1$ individual players. The players are given $n$ arms, enumerated by $[n] := \{1, 2, \ldots, n\}$. Each arm $i \in [n]$ is associated with a reward, which is a $[0, 1]$-valued random variable with expectation $p_i$. For convenience, we assume that the arms are ordered by their expected rewards, that is $p_1 \geq p_2 \geq \cdots \geq p_n$. At every time step $t = 1, 2, \ldots, T$, each player pulls one arm of his choice and observes an independent sample of its reward. Each player may choose any of the arms, regardless of the other players and their actions. At the end of the game, each player must commit to a single arm. In a *communication round*, that may take place at any predefined time step, each player may broadcast a message to all other players. While we do not restrict the size of each message, in a reasonable implementation a message should not be larger than $\tilde{O}(n)$ bits.

In the *best-arm identification* version of the problem, the goal of a multi-player algorithm given some target confidence level $\delta > 0$, is that with probability at least $1 - \delta$ *all* players correctly identify the best arm (i.e. the arm having the maximal expected reward). For simplicity, we assume in this setting that the best arm is unique. Similarly, in the $(\varepsilon, \delta)$-PAC variant the goal is that each player finds an $\varepsilon$-optimal (or "$\varepsilon$-best") arm, that is an arm $i$ with $p_i \geq p_1 - \varepsilon$, with high probability. In this paper we focus on the more general $(\varepsilon, \delta)$-PAC setup, which also includes best-arm identification for $\varepsilon = 0$.

We use the notation $\Delta_i := p_1 - p_i$ to denote the suboptimality gap of arm $i$, and occasionally use $\Delta_\star := \Delta_2$ for denoting the minimal gap. In the best-arm version of the problem, where we assume that the best arm is unique, we have $\Delta_i > 0$ for all $i > 1$. When dealing with the $(\varepsilon, \delta)$-PAC setup, we also consider the truncated gaps $\Delta_i^\varepsilon := \max\{\Delta_i, \varepsilon\}$. In the context of MAB problems, we are interested in deriving distribution-dependent bounds, namely, bounds that are stated as a function of $\varepsilon, \delta$ and also the distribution-specific values $\Delta := (\Delta_2, \ldots, \Delta_n)$. The $\tilde{O}$ notation in our bounds hides polylogarithmic factors in $n, k, \varepsilon, \delta$, and also in $\Delta_2, \ldots, \Delta_n$. In the case of serial exploration algorithms (i.e., when there is only one player), the lower bounds of Mannor and Tsitsiklis [20] and Audibert et al. [3] show that in general $\tilde{\Omega}(H_\varepsilon)$ pulls are necessary for identifying an $\varepsilon$-arm, where

$$H_\varepsilon := \sum_{i=2}^{n} \frac{1}{(\Delta_i^\varepsilon)^2} \,. \tag{1}$$

Intuitively, the hardness of the task is therefore captured by the quantity $H_\varepsilon$, which is roughly the number of arm pulls needed to find an $\varepsilon$-best arm with a reasonable probability; see also [3] for a discussion. Our goal in this work is therefore to establish bounds in the distributed model that are expressed as a function of $H_\varepsilon$, in the same vein of the bounds known in the classic MAB setup.

### 2.1 Baseline approaches

We now discuss several baseline approaches for the problem, starting with our main focus—the single round setting. The first obvious approach, already mentioned earlier, is the *no-communication* strategy: just let each player explore the arms in isolation of the other players, following an independent instance of some serial strategy; at the end of the executions, all players hold an $\varepsilon$-best arm. Clearly, this approach performs poorly in terms of learning performance, needing $\tilde{\Omega}(H_\varepsilon)$ pulls per player in the worst case and not leading to any parallel speed-up.

Another straightforward approach is to employ a *majority vote* among the players: let each player independently identify an arm, and choose the arm having most of the votes (alternatively, at least half of the votes). However, this approach does not lead to any improvement in performance: for this vote to work, each player has to solve the problem correctly with reasonable probability, which already require $\tilde{\Omega}(H_\varepsilon)$ pulls of each. Even if we somehow split the arms between players and let each player explore a share of them, a majority vote would still fail since those players getting the "good" arms might have to pull arms $\tilde{\Omega}(H_\varepsilon)$ times—a small MAB instance might be as hard as the full-sized problem (in terms of the complexity measure $H_\varepsilon$).

When considering algorithms employing multiple communication rounds, we use an ideal *simulated serial* algorithm (i.e., a full-communication approach) as our baseline. This approach is of course prohibited in our context, but is able to achieve the optimal parallel speed-up, linear in the number of players $k$.

## 2.2 Our results

We now discuss our approach and overview our algorithmic results. These are summarized in Table 1 below, that compares the different algorithms in terms of parallel speed-up and communication.

Our approach for the one-round case is based on the idea of majority vote. For the best-arm identification task, our observation is that by letting each player explore a smaller set of $n/\sqrt{k}$ arms chosen at random and choose one of them as "best", about $\sqrt{k}$ of the players would come up with the *global* best arm. This (partial) consensus on a single arm is a key aspect in our approach, since it allows the players to identify the correct best arm among the votes of all $k$ players, after sharing information only once. Our approach leads to a factor $\sqrt{k}$ parallel speed-up which, as we demonstrate in our lower bound, is the optimal factor in this setting. Although our goal here is pure exploration, in our algorithms each player follows an explore-exploit strategy. The idea is that a player should sample his recommended arm as much as his budget permits, even if it was easy to identify in his small-sized problem. This way we can guarantee that the top arms are sampled to a sufficient precision by the time each of the players has to choose a single best arm.

The algorithm for the $(\varepsilon, \delta)$-PAC setup is similar, but its analysis is more challenging. As mentioned above, an agreement on a single arm is essential for a vote to work. Here, however, there might be several $\varepsilon$-best arms, so arriving at a consensus on a single one is more difficult. Nonetheless, by examining two different regimes, namely when there are "many" $\varepsilon$-best arms and when there are "few" of them, our analysis shows that a vote can still work and achieve the $\sqrt{k}$ multiplicative speed-up.

In the case of multiple communication rounds, we present a distributed elimination-based algorithm that discards arms right after each communication round. Between rounds, we share the work load between players uniformly. We show that the number of such rounds can be reduced to as low as $O(\log(1/\varepsilon))$, by eliminating all $2^{-r}$-suboptimal arms in the $r$'th round. A similar idea was employed in [4] for improving the regret bound of UCB with respect to the parameters $\Delta_i$. We also use this technique to develop an algorithm that performs only $R$ communication rounds, for any given parameter $R \geq 1$, that achieves a slightly worse multiplicative $\varepsilon^{2/R}k$ speed-up.

| SETTING | ALGORITHM | SPEED-UP | COMMUNICATION |
|---------|-----------|----------|---------------|
| ONE-ROUND | No-Communication | 1 | none |
| | Majority Vote | 1 | 1 round |
| | Algorithm 1,2 | $\sqrt{k}$ | 1 round |
| MULTI-ROUND | Serial (simulated) | $k$ | every time step |
| | Algorithm 3 | $k$ | $O(\log \frac{1}{\varepsilon})$ rounds |
| | Algorithm 3' | $\varepsilon^{2/R} \cdot k$ | $R$ rounds |

Table 1: Summary of baseline approaches and our results. The speed-up results are asymptotic (logarithmic factors are omitted).

## 3 One Communication Round

This section considers the most basic variant of the multi-player MAB problem, where each player is only allowed a single transmission, when finishing her queries. For the clarity of exposition, we first consider the best-arm identification setting in Section 3.1. Section 3.2 deals with the $(\varepsilon, \delta)$-PAC setup. We demonstrate the tightness of our result in Section 3.3 with a lower bound for the required budget of arm pulls in this setting.

Our algorithms in this section assume the availability of a serial algorithm $\mathcal{A}(A, \varepsilon)$, that given a set of arms $A$ and target accuracy $\varepsilon$, identifies an $\varepsilon$-best arm in $A$ with probability at least $2/3$ using no more than

$$c_{\mathcal{A}} \sum_{i \in A} \frac{1}{(\Delta_i^{\varepsilon})^2} \log \frac{|A|}{\Delta_i^{\varepsilon}} \tag{2}$$

arm pulls, for some constant $c_{\mathcal{A}} > 1$. For example, the Successive Elimination algorithm [14] and the Exp-Gap Elimination algorithm [18] provide a guarantee of this form. Essentially, any exploration strategy whose guarantee is expressed as a function of $H_\varepsilon$ can be used as the procedure $\mathcal{A}$, with technical modifications in our analysis.

### 3.1 Best-arm Identification Algorithm

We now describe our one-round best-arm identification algorithm. For simplicity, we present a version matching $\delta = 1/3$, meaning that the algorithm produces the correct arm with probability at least $2/3$; we later explain how to extend it to deal with arbitrary values of $\delta$.

Our algorithm is akin to a majority vote among the multiple players, in which each player pulls arms in two stages. In the first EXPLORE stage, each player independently solves a "smaller" MAB instance on a random subset of the arms using the exploration strategy $\mathcal{A}$. In the second EXPLOIT stage, each player exploits the arm identified as "best" in the first stage, and communicates that arm and its observed average reward. See Algorithm 1 below for a precise description. An appealing feature of our algorithm is that it requires each player to transmit a single message of constant size (up to logarithmic factors).

In Theorem 3.1 we prove that Algorithm 1 indeed achieves the promised upper bound.

**Theorem 3.1.** *Algorithm 1 identifies the best arm correctly with probability at least $2/3$ using no more than*

$$O\left( \frac{1}{\sqrt{k}} \cdot \sum_{i=2}^{n} \frac{1}{\Delta_i^2} \log \frac{n}{\Delta_i} \right)$$

*arm pulls per player, provided that $6 \le \sqrt{k} \le n$. The algorithm uses a single communication round, in which each player communicates $\tilde{O}(1)$ bits.*

By repeating the algorithm $O(\log(1/\delta))$ times and taking the majority vote of the independent runs, we can amplify the success probability to $1 - \delta$ for any given $\delta > 0$. Note that we can still do that with one communication round (at the end of all executions), but each player now has to communicate $O(\log(1/\delta))$ values[1].

**Theorem 3.2.** *There exists a $k$-player algorithm that given $\tilde{O}\left( \frac{1}{\sqrt{k}} \sum_{i=2}^{n} 1/\Delta_i^2 \right)$ arm pulls, identifies the best arm correctly with probability at least $1 - \delta$. The algorithm uses a single communication round, in which each player communicates $O(\log(1/\delta))$ numerical values.*

---

**Algorithm 1** ONE-ROUND BEST-ARM

**input** time horizon $T$
**output** an arm
1: **for** player $j = 1$ to $k$ **do**
2:     choose a subset $A_j$ of $6n/\sqrt{k}$ arms uniformly at random
3:     EXPLORE: execute $i_j \leftarrow \mathcal{A}(A_j, 0)$ using at most $\frac{1}{2}T$ pulls (and halting the algorithm early if necessary);
        if the algorithm fails to identify any arm or does not terminate gracefully, let $i_j$ be an arbitrary arm
4:     EXPLOIT: pull arm $i_j$ for $\frac{1}{2}T$ times and let $\hat{q}_j$ be its average reward
5:     communicate the numbers $i_j, \hat{q}_j$
6: **end for**
7: let $k_i$ be the number of players $j$ with $i_j = i$, and define $A = \{i : k_i > \sqrt{k}\}$
8: let $\hat{p}_i = (1/k_i) \sum_{\{j : i_j = i\}} \hat{q}_j$ for all $i$
9: **return** $\arg\max_{i \in A} \hat{p}_i$; if the set $A$ is empty, output an arbitrary arm.

---

We now prove Theorem 3.1. We show that a budget $T$ of samples (arm pulls) per player, where

$$T \ge \frac{24 c_{\mathcal{A}}}{\sqrt{k}} \cdot \sum_{i=2}^{n} \frac{1}{\Delta_i^2} \ln \frac{n}{\Delta_i} \,, \tag{3}$$

suffices for the players to jointly identify the best arm $i^\star$ with the desired probability. Clearly, this would imply the bound stated in Theorem 3.1. We note that we did not try to optimize the constants in the above expression.

We begin by analyzing the EXPLORE phase of the algorithm. Our first lemma shows that each player chooses the global best arm and identifies it as the local best arm with sufficiently large probability.

**Lemma 3.3.** *When* (3) *holds, each player identifies the (global) best arm correctly after the* EX-PLORE *phase with probability at least* $2/\sqrt{k}$.

We next address the EXPLOIT phase. The next simple lemma shows that the popular arms (i.e. those selected by many players) are estimated to a sufficient precision.

**Lemma 3.4.** *Provided that* (3) *holds, we have* $|\hat{p}_i - p_i| \leq \frac{1}{2}\Delta_\star$ *for all arms* $i \in A$ *with probability at least* $5/6$.

Due to lack of space, the proofs of the above lemmas are omitted and can be found in [16]. We can now prove Theorem 3.1.

*Proof (of Theorem 3.1).* Let us first show that with probability at least $5/6$, the best arm $i$ is contained in the set $A$. To this end, notice that $k_{i^\star}$ is the sum of $k$ i.i.d. Bernoulli random variables $\{I_j\}_j$ where $I_j$ is the indicator of whether player $j$ chooses arm $i^\star$ after the EXPLORE phase. By Lemma 3.3 we have that $\mathbf{E}[I_j] \geq 2/\sqrt{k}$ for all $j$, hence by Hoeffding's inequality, $\Pr[k_{i^\star} \leq \sqrt{k}] \leq \Pr[k_{i^\star} - \mathbf{E}[k_{i^\star}] \leq -\sqrt{k}] \leq \exp(-2k/k) \leq 1/6$ which implies that $i^\star \in A$ with probability at least $5/6$.

Next, note that with probability at least $5/6$ the arm $i \in A$ having the highest empirical reward $\hat{p}_i$ is the one with the highest expected reward $p_i$. Indeed, this follows directly from Lemma 3.4 that shows that with probability at least $5/6$, for all arms $i \in A$ the estimate $\hat{p}_i$ is within $\frac{1}{2}\Delta$ of the true bias $p_i$. Hence, via a union bound we conclude that with probability at least $2/3$, the best arm is in $A$ and has the highest empirical reward. In other words, with probability at least $2/3$ the algorithm outputs the best arm $i^\star$. □

## 3.2 $(\varepsilon, \delta)$-PAC Algorithm

We now present an algorithm whose purpose is to recover an $\varepsilon$-optimal arm. Here, there might be more than one $\varepsilon$-best arm, so each "successful" player might come up with a different $\varepsilon$-best arm. Nevertheless, our analysis below shows that with high probability, a subset of the players can still agree on a single $\varepsilon$-best arm, which makes it possible to identify it among the votes of all players. Our algorithm is described in Algorithm 2, and the following theorem states its guarantees.

**Theorem 3.5.** *Algorithm 2 identifies a* $2\varepsilon$-*best arm with probability at least* $2/3$ *using no more than*

$$O\left(\frac{1}{\sqrt{k}} \cdot \sum_{i=2}^{n} \frac{1}{(\Delta_i^\varepsilon)^2} \log \frac{n}{\Delta_i^\varepsilon}\right)$$

*arm pulls per player, provided that* $24 \leq \sqrt{k} \leq n$. *The algorithm uses a single communication round, in which each player communicates* $\tilde{O}(1)$ *bits.*

Before proving the theorem, we first state several key lemmas. In the following, let $n_\varepsilon$ and $n_{2\varepsilon}$ denote the number of $\varepsilon$-best and $2\varepsilon$-best arms respectively. Our analysis considers two different regimes: $n_{2\varepsilon} \leq \frac{1}{50}\sqrt{k}$ and $n_{2\varepsilon} > \frac{1}{50}\sqrt{k}$, and shows that in any case,

$$T \geq \frac{400c_{\mathcal{A}}}{\sqrt{k}} \sum_{i=2}^{n} \frac{1}{(\Delta_i^\varepsilon)^2} \ln \frac{24n}{\Delta_i^\varepsilon} \tag{4}$$

suffices for identifying a $2\varepsilon$-best arm with the desired probability. Clearly, this implies the bound stated in Theorem 3.5.

The first lemma shows that at least one of the players is able to find an $\varepsilon$-best arm. As we later show, this is sufficient for the success of the algorithm in case there are many $2\varepsilon$-best arms.

**Lemma 3.6.** *When* (4) *holds, at least one player successfully identifies an* $\varepsilon$-*best arm in the* EX-PLORE *phase, with probability at least* $5/6$.

The next lemma is more refined and states that in case there are few $2\varepsilon$-best arms, the probability of each player to successfully identify an $\varepsilon$-best arm grows linearly with $n_\varepsilon$.

**Lemma 3.7.** *Assume that* $n_{2\varepsilon} \leq \frac{1}{50}\sqrt{k}$. *When* (4) *holds, each player identifies an* $\varepsilon$-*best arm in the* EXPLORE *phase, with probability at least* $2n_\varepsilon/\sqrt{k}$.

The last lemma we need analyzes the accuracy of the estimated rewards of arms in the set $A$.

**Lemma 3.8.** *With probability at least $5/6$, we have $|\hat{p}_i - p_i| \leq \varepsilon/2$ for all arms $i \in A$.*

For the proofs of the above lemmas, refer to [16]. We now turn to prove Theorem 3.5.

*Proof.* We shall prove that with probability $5/6$ the set $A$ contains at least one $\varepsilon$-best arm. This would complete the proof, since Lemma 3.8 assures that with probability $5/6$, the estimates $\hat{p}_i$ of all arms $i \in A$ are at most $\varepsilon/2$-away from the true reward $p_i$, and in turn implies (via a union bound) that with probability $2/3$ the arm $i \in A$ having the maximal empirical reward $\hat{p}_i$ must be a $2\varepsilon$-best arm.

First, consider the case $n_{2\varepsilon} > \frac{1}{50}\sqrt{k}$. Lemma 3.6 shows that with probability $5/6$ there exists a player $j$ that identifies an $\varepsilon$-best arm $i_j$. Since for at least $n_{2\varepsilon}$ arms $\Delta_i \leq 2\varepsilon$, we have

$$t_{i_j} \geq \tfrac{1}{2}T \geq \frac{400}{2\sqrt{k}} \cdot \frac{n_{2\varepsilon} - 1}{(2\varepsilon)^2} \ln \frac{24n}{2\varepsilon}$$

$$\geq \frac{1}{\varepsilon^2} \ln(12n) \,,$$

that is, $i_j \in A$.

---

**Algorithm 2** ONE-ROUND $\varepsilon$-ARM

**input** time horizon $T$, accuracy $\varepsilon$
**output** an arm
 1: **for** player $j = 1$ to $k$ **do**
 2:     choose a subset $A_j$ of $12n/\sqrt{k}$ arms uniformly at random
 3:     EXPLORE: execute $i_j \leftarrow \mathcal{A}(A_j, \varepsilon)$ using at most $\frac{1}{2}T$ pulls (and halting the algorithm early if necessary);
    if the algorithm fails to identify any arm or does not terminate gracefully, let $i_j$ be an arbitrary arm
 4:     EXPLOIT: pull arm $i_j$ for $\frac{1}{2}T$ times, and let $\hat{q}_j$ be the average reward
 5:     communicate the numbers $i_j, \hat{q}_j$
 6: **end for**
 7: let $k_i$ be the number of players $j$ with $i_j = i$
 8: let $t_i = \frac{1}{2}k_i T$ and $\hat{p}_i = (1/k_i)\sum_{\{j\,:\,i_j=i\}} \hat{q}_j$ for all $i$
 9: define $A = \{i \in [n] : t_i \geq (1/\varepsilon^2)\ln(12n)\}$
10: **return** $\arg\max_{i \in A} \hat{p}_i$; if the set $A$ is empty, output an arbitrary arm.

---

Next, consider the case $n_{2\varepsilon} \leq \frac{1}{50}\sqrt{k}$. Let $N$ denote the number of players that identified some $\varepsilon$-best arm. The random variable $N$ is a sum of Bernoulli random variables $\{I_j\}_j$ where $I_j$ indicates whether player $j$ identified some $\varepsilon$-best arm. By Lemma 3.7, $\mathbf{E}[I_j] \geq 2n_\varepsilon/\sqrt{k}$ and thus by Hoeffding's inequality, $\Pr[N < n_\varepsilon\sqrt{k}] = \Pr[N - \mathbf{E}[N] \leq -n_\varepsilon\sqrt{k}] \leq \exp(-2n_\varepsilon^2) \leq 1/6$. That is, with probability $5/6$, at least $n_\varepsilon\sqrt{k}$ players found an $\varepsilon$-best arm. A pigeon-hole argument now shows that in this case there exists an $\varepsilon$-best arm $i^\star$ selected by at least $\sqrt{k}$ players. Hence, with probability $5/6$ the number of samples of this arm collected in the EXPLOIT phase is at least $t_{i^\star} \geq \sqrt{k}T/2 > (1/\varepsilon^2)\ln(12n)$, which means that $i^\star \in A$. $\qquad\square$

### 3.3 Lower Bound

The following theorem suggests that in general, for identifying the best arm $k$ players achieve a multiplicative speed-up of at most $\tilde{O}(\sqrt{k})$ when allowing one transmission per player (at the end of the game). Clearly, this also implies that a similar lower bound holds in the PAC setup, and proves that our algorithmic results for the one-round case are essentially tight.

**Theorem 3.9.** *For any $k$-player strategy that uses a single round of communication, there exist rewards $p_1, \ldots, p_n \in [0, 1]$ and integer $T$ such that*

- *each individual player must use at least $T/\sqrt{k}$ arm pulls for them to collectively identify the best arm with probability at least $2/3$;*
- *there exist a single-player algorithm that needs at most $\tilde{O}(T)$ pulls for identifying the best arm with probability at least $2/3$.*

The proof of the theorem is omitted due to space constraints and can be found in [16].

## 4 Multiple Communication Rounds

In this section we establish an explicit tradeoff between the performance of a multi-player algorithm and the number of communication rounds it uses, in terms of the accuracy $\varepsilon$. Our observation is that

by allowing $O(\log(1/\varepsilon))$ rounds of communication, it is possible to achieve the optimal speedup of factor $k$. That is, we do not gain any improvement in learning performance by allowing more than $O(\log(1/\varepsilon))$ rounds.

Our algorithm is given in Algorithm 3. The idea is to eliminate in each round $r$ (i.e., right after the $r$th communication round) all $2^{-r}$-suboptimal arms. We accomplish this by letting each player sample uniformly all remaining arms and communicate the results to other players. Then, players are able to eliminate suboptimal arms with high confidence. If each such round is successful, after $\log_2(1/\varepsilon)$ rounds only $\varepsilon$-best arms survive. Theorem 4.1 below bounds the number of arm pulls used by this algorithm (a proof can be found in [16]).

**Theorem 4.1.** *With probability at least $1 - \delta$, Algorithm 3*

- *identifies the optimal arm using*

$$O\left(\frac{1}{k} \cdot \sum_{i=2}^{n} \frac{1}{(\Delta_i^\varepsilon)^2} \log\left(\frac{n}{\delta} \log \frac{1}{\Delta_i^\varepsilon}\right)\right)$$

  *arm pulls per player;*

---

**Algorithm 3** MULTI-ROUND $\varepsilon$-ARM

**input** $(\varepsilon, \delta)$
**output** an arm
1: initialize $S_0 \leftarrow [n], r \leftarrow 0, t_0 \leftarrow 0$
2: **repeat**
3:     set $r \leftarrow r + 1$
4:     let $\varepsilon_r \leftarrow 2^{-r}, t_r \leftarrow (2/k\varepsilon_r^2) \ln(4nr^2/\delta)$
5:     **for** player $j = 1$ to $k$ **do**
6:         sample each arm $i \in S_{r-1}$ for $t_r - t_{r-1}$ times
7:         let $\hat{p}_{j,i}^r$ be the average reward of arm $i$ (in all rounds so far of player $j$)
8:         communicate the numbers $\hat{p}_{j,1}^r, \ldots, \hat{p}_{j,n}^r$
9:     **end for**
10:    let $\hat{p}_i^r = (1/k) \sum_{j=1}^{k} \hat{p}_{j,i}^r$ for all $i \in S_{r-1}$, and let $\hat{p}_\star^r = \max_{i \in S_{r-1}} \hat{p}_i^r$
11:    set $S_r \leftarrow S_{r-1} \setminus \{i \in S_{r-1} : \hat{p}_i^r < \hat{p}_\star^r - \varepsilon_r\}$
12: **until** $\varepsilon_r \leq \varepsilon/2$ or $|S_r| = 1$
13: **return** an arm from $S_r$

---

- *terminates after at most $1 + \lceil \log_2(1/\varepsilon) \rceil$ rounds of communication (or after $1 + \lceil \log_2(1/\Delta_\star) \rceil$ rounds for $\varepsilon = 0$).*

By properly tuning the elimination thresholds $\varepsilon_r$ of Algorithm 3 in accordance with the target accuracy $\varepsilon$, we can establish an explicit trade-off between the number of communication rounds and the number of arm pulls each player needs. In particular, we can design a multi-player algorithm that terminates after at most $R$ communication rounds, for any given parameter $R > 0$. This, however, comes at the cost of a compromise in learning performance as quantified in the following corollary.

**Corollary 4.2.** *Given a parameter $R > 0$, set $\varepsilon_r \leftarrow \varepsilon^{r/R}$ for all $r \geq 1$ in Algorithm 3. With probability at least $1 - \delta$, the modified algorithm*

- *identifies an $\varepsilon$-best arm using $\tilde{O}((\varepsilon^{-2/R}/k) \cdot \sum_{i=2}^{n} (1/\Delta_i^\varepsilon)^2)$ arm pulls per player;*
- *terminates after at most $R$ rounds of communication.*

## 5 Conclusions and Further Research

We have considered a collaborative MAB exploration problem, in which several independent players explore a set of arms with a common goal, and obtained the first non-trivial results in such setting. Our main results apply for the specifically interesting regime where each of the players is allowed a single transmission; this setting fits naturally to common distributed frameworks such as MapReduce. An interesting open question in this context is whether one can obtain a strictly better speed-up result (which, in particular, is independent of $\varepsilon$) by allowing more than a single round. Even when allowing merely two communication rounds, it is unclear whether the $\sqrt{k}$ speed-up can be improved. Intuitively, the difficulty here is that in the second phase of a reasonable strategy each player should focus on the arms that excelled in the first phase; this makes the sub-problems being faced in the second phase as hard as the entire MAB instance, in terms of the quantity $H_\varepsilon$. Nevertheless, we expect our one-round approach to serve as a building-block in the design of future distributed exploration algorithms, that are applicable in more complex communication models.

An additional interesting problem for future research is how to translate our results to the regret minimization setting. In particular, it would be nice to see a conversion of algorithms like UCB [5] to a distributed setting. In this respect, perhaps a more natural distributed model is a one resembling that of Kanade et al. [17], that have established a regret vs. communication trade-off in the non-stochastic setting.

## Footnotes

[1]In fact, by letting each player pick a slightly larger subset of $O(\sqrt{\log(1/\delta)} \cdot n/\sqrt{k})$ arms, we can amplify the success probability to $1 - \delta$ without needing to communicate more than 2 values per player. However, this approach only works when $k = \Omega(\log(1/\delta))$.

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
