[Reviews · NeurIPS 2013]

Submitted by Assigned_Reviewer_5

The paper considers the possibility of parallelizing the best arm identification problem. In specific, they ask how and what speed-up can by achieved by running k instances of an algorithm in parallel if they are allowed to share some information with each other. The information sharing is done by each of the instances broadcasting some information for the rest of the algorithms (typically at the end of the run), but with the restriction that the number of broadcasted bits is limited (typically of size \tilde{O}(n) for each instance).

The authors compare their algorithms to the full-communication method, the no-communication method, and the method that takes the majority vote (MV) of the individuals recommendations. The first one generates too much network data, the second one obviously achieves no speed-up, and, as they claim, the MV method achieves no speed-up either. In contrast, the algorithm they propose achieves a speed-up of \sqrt{k}. They also show that this is optimal if only one round of communication is allowed.

Finally, they show that if more communication rounds are allowed, then there is a trade-off between the possible speed-up (up to k) and the amount of information exchanged.



EVALUATIION

The writing style is clear (mostly---but more about that below), the proposed solutions are simple but clever, and the problem is both non-trivial and reasonable (their motivating example is the distributed computation model of MapReduce).

However, there are two issues which should be clarified.

First of all, please define precisely what you mean by ``speed-up'', and state your goals formally! It is quite natural to think that if some algorithm A achieves confidence 1-\delta in time T, whereas a distributed version achieves confidence 1-\delta' during the same time for some \delta' much less than \delta, then this is a kind of speed-up. (Indeed, for a stand-alone version of A to achieve the same 1-\delta' confidence could take much longer than T.) And, in this sense, MV would clearly achieve a speed-up: if time T is enough for A to output the best arm with probability 2/3, then increasing k to infinity, and simply taking the majority vote of the k instances of A (each run for time T) would return the best arm with a probability getting arbitrarily close to 1.

Also, the performance of the MV method should be treated with more care. In order to achieve that (in expectation) at least half of the players return the best arm, the individual algorithms indeed have to output the best arm with probability at least 1/2. However, for the MV method to work this is not at all required. In fact, it is enough to have that the individual algorithms output the best arm with higher probability as any other suboptimal arm.


Finally, two less significant questions:
1. In the second line of the proof of Lemma 3.3 why do you have the multiplicator 6 in front of H/\sqrt{k}?
2. In the second line of the proof of Lemma 3.4 shouldn't 12n be devided by \Delta^\star in the logarithm?
Summary: The problem is both non-trivial and reasonable, the proposed solutions are simple
but clever, and the writing style is mostly clear. Some issues are raised though
(regarding how they define their ``speed-up'' and why they claim that the majority
vote does not work) that require clarifications.

Submitted by Assigned_Reviewer_6

Paper summary:
--------------
The paper presents algorithms and their analysis for several settings of distributed best-arm identification in multiarmed bandits (MAB) problem. The authors consider best-arm and epsilon-best arm identification by k "workers" with one and with R communication rounds. For one round of communication the authors demonstrate \sqrt{k} speed-up and for R rounds of communication the authors demonstrate \epsilon^{2/R} k speed-up. In particular, for R = O(log(1/\epsilon)) the algorithm achieves the best-possible speed-up factor of k. The result for one communication round is accompanied by a matching lower bound.

Quality, clarity, originality, and significance:
------------------------------------------------
The paper is of good quality, clearly written, significant, and original. (I did not check the supplementary material, except for the proof of the lower bound, but other results sound plausible).
Summary: I think this is a good paper that would be interesting to the NIPS community. My comments are all minor and can be easily corrected by the authors.

Submitted by Assigned_Reviewer_8

The paper considers the problem of minimizing the simple regret in multi-armed bandit problems, that is, it is concerned with the sample complexity of identifying the arm with highest payoff. The authors aim to find out whether it is possible to speed up previously known algorithms by parallel processing on k processors. They propose an algorithm that attains a sqrt(k) speedup by letting the nodes communicate only once, and they show that no greater speedup can be achieved when restricting communication to a single round. Furthermore, it is shown that it is possible to attain the optimal speedup of k by allowing O(log 1/epsilon) communication rounds, where epsilon is the desired precision.

I have greatly enjoyed reading the paper; it is very well-written. Even though it seemed to me that every interesting problem has already been solved concerning the problem of best-arm selection, the authors manage to come up with a novel, practically relevant framework, and prove elegant results concerning the complexity of the resulting learning problem. The results are quite original, even though proving them did not need any spectacular new techniques. Nevertheless, the technical content seems sound. Overall, I have only very minor complaints/additional questions concerning the paper.


Detailed comments
-------------------------
182 The results concerning using Alg 3 for selecting the best arm are not communicated appropriately. You only state your results for epsilon > 0, these should be also stated in terms of Delta_* for completeness.
194 In the last row, r -> R to stay consistent with Corollary 4.2
211 Wouldn’t it be more straightforward to require that an algorithm successfully returns the best arm with probability at least 1-delta? Kalyanakrishnan et al. (2012) and Gabillon et al. (2012) provide such algorithms for the more general problem of selecting the m best arms. Any comment on how your results generalize to that setting?
263 The wording of Lemma 3.3 is somewhat confusing: as can be deducted from the proof of Lemma 3.4, it should say that the probability that a fixed player identifies the best arm with such and such probability. Currently, the statement can be misunderstood as if it was saying that *all* players identify the correct arm with said probability.
307 Even though I understand that you did not try to optimize the constants, can you give any insights as to how much these factors can be possibly improved? These factors of 400 and 50 seem to come out of nowhere, is it possible to significantly reduce them?
398 Delta_*
467 H. D. III -> H. Daume III, or, even better, H. Daumé III
Summary: The paper provides an elegant solution to a well-motivated new learning problem. The writing style and the technical quality is excellent.
Author Feedback

Author rebuttal: We thank the reviewers for careful reading and thoughtful comments.

Reviewer_5:
=========
** Speed-up in terms of delta: Note that for evaluating speed-ups and comparing between different approaches we compare bounds over the required time for *constant* success prob. (say, delta=2/3). There is indeed a subtle trade-off between delta and the other parameters of the problem (T, k) that we wish to avoid this way, so that comparing bounds becomes easier and meaningful. We will highlight this point in the final version.
** Analysis of majority vote: In fact, for MV to work each individual player has to output the correct arm with probability which is significantly larger than of other arms. Otherwise, for instance when the success probability of each player is 1/n+1/k, it is very probable that some suboptimal arm would get the largest number of votes. This is also demonstrated by the instance we consider in our lower-bound: it is not too difficult to show that in that case, MV does not achieve any speed-up. We will elaborate a bit on this and add references in the final version.

** Proof of Lemma 3.3: Note that each player chooses 6*n/sqrt{k} arms at random, thus the expectation of H_j is 6/sqrt{k} * H (conditioned on the event i^* \in A_j, as appears in the proof).
** Proof of Lemma 3.4: We did (intentionally) miss Delta^* from the log factor, while keeping the inequality valid: in fact, this loose bound suffices for the analysis of the exploit phase.

Reviewer_6:
=========
** L103: You are correct, our results apply to any distributions supported in the interval [0,1], not just binary rewards. We will modify the presentation to highlight this.
** L344: Right, we missed a log(1/eps) factor here (the inequality is still true for eps<1, though).

Reviewer_8:
=========
** Requirement of success prob. >= 2/3 (and not 1-delta) for strategy A: We chose to present our assumptions this way since this is precisely what our algorithm and analysis actually require. We will consider adopting a more 'standard' presentation of 1-delta prob. (along with a small comment) as per your suggestion.
** Extension to m-best-arm selection: We did not consider this extension previously. However, it seems that the solution should not be too different from that of the best arm identification.
** Large constants: we believe these are just artifacts of our analysis and the 'true' constants are much smaller. One way to improve the constants in our bounds is by trying to avoid Markov's inequality altogether, but this would require analyzing dependencies between the random variables denoting the gaps (between arms) faced by each player - which might be a non-trivial task (that would not lead to any asymptotic improvement).